# Can Crude Oil Exploration Influence the Phytochemicals and Bioactivity of Medicinal Plants? A Case of Nigerian *Vernonia amygdalina* and *Ocimum gratissimum*

**DOI:** 10.3390/molecules27238372

**Published:** 2022-11-30

**Authors:** Oluwatofunmilayo A. Diyaolu, Emmanuel T. Oluwabusola, Alfred F. Attah, Eric O. Olori, Adeshola A. Fagbemi, Gagan Preet, Sylvia Soldatou, Jones O. Moody, Marcel Jaspars, Rainer Ebel

**Affiliations:** 1Marine Biodiscovery Centre, Department of Chemistry, University of Aberdeen, Aberdeen AB24 3UE, UK; 2Department of Pharmacognosy and Drug Development, Faculty of Pharmaceutical Sciences, University of Ilorin, Ilorin 240003, Nigeria; 3Department of Pharmaceutical Chemistry, Faculty of Pharmacy, University of Ibadan, Ibadan 200005, Nigeria; 4Department of Pharmaceutical Chemistry, Faculty of Pharmacy, Lead City University, Ibadan 200255, Nigeria; 5Department of Pharmacognosy, Faculty of Pharmacy, University of Ibadan, Ibadan 200005, Nigeria

**Keywords:** *Vernonia amygdalina* leaves, *Ocimum gratissimum* leaves, UPLC-QTOF-MS, sickle cell anaemia, molecular docking, metabolomics, pharmacophore, drug design, in silico

## Abstract

The Nigerian Niger-Delta crude oil exploration often results in spills that affect indigenous medicinal plant biodiversity, likely changing the phytochemical profile of surviving species, their bioactivity or toxicity. In crude oil-rich Kokori and crude oil-free Abraka, classic examples of indigenous plants occupying the medicine-food interface include *Vernonia amygdalina* (VAL) and *Ocimum gratissimum* leaves (OGL). These plants are frequently utilised during pregnancy and in anaemia. To date, no scientific investigation has been reported on the potential changes to the phytochemical or bioactivity of the study plants. To discuss the similarities and dissimilarities in antisickling bioactivity and phytochemicals in VAL and OGL collected from Kokori (VAL-KK and OGL-KK) and Abraka (VAL-AB and OGL-AB), in silico, in vitro and comparative UPLC-QTOF-MS analysis was performed. Nine unique compounds were identified in OGL-KK, which have never been reported in the literature, while differences in antisickling potentials were observed in VAL-KK, OGL-KK and, VAL-AB, OGL-AB. Our findings show that VAL-AB and OGL-AB are richer and more diverse in phytochemicals and displayed a slightly higher antisickling activity than VAL-KK and OGL-KK. Ligand-based pharmacophore modelling was performed to understand the potential compounds better; this study may provide a basis for explaining the effect of crude oil spills on secondary metabolites and a reference for further research.

## 1. Introduction

The World Health Organisation has reported that no less than 80% of the African populace relies on medicinal herbs for their essential healthcare needs. However, ethnomedicinal plant use is partly tied to poverty, cultural resistance and poor accessibility to modern healthcare [1]. For example, *Vernonia amygdalina* leaves (VAL) and *Ocimum gratissimum* leaves (OGL) are commonly used to manage sickle cell disease (SCD) [2]. The leaves are macerated in an aqueous or alcoholic medium, and the extracted juice is administered [3,4]. *Vernonia amygdalina*, commonly known as “bitter leaf,” is a small shrub with dark green leaves and rough bark [5], mainly growing in tropical Africa but domesticated in many parts of West Africa. *Ocimum gratissimum*, also known as African basil, is cultivated in many gardens around village huts in Nigeria for medicinal and culinary uses [6]. Both VAL and OGL contain a myriad of phytochemicals. More than forty compounds belonging to several classes of secondary metabolites with differing bioactivities have been isolated and characterised from VAL and OGL. The isolated compounds have shown a broad spectrum of activities ranging from antifeedant [7], antischistosomal [8], antiplasmodial [9], antioxidant [10], anti-inflammatory [11] and anticancer activities [12].

VAL and OGL are both found in nature close to rivers and lakes. They can thrive on all soil types but grow better on humus-rich soils. However, they can withstand drought and other harsh conditions because they have continuously survived the massive crude oil spillage in Kokori Delta State, Nigeria. Kokori is crude oil-rich and one such community in Delta State where crude oil exploration and effluent release are common, and these activities have led to the death of many medicinal plants. However, VAL and OGL have survived with a possibility of an alteration of the phytochemical profile. The indigenes of this region continue to use these plants for medicinal purposes without considering the potentially detrimental effects this may have. Although there have been reports on the heavy metal content and proximate analysis of VAL and OGL in crude oil-rich communities in Delta state [13], there has been a lack of profound reports on the comprehensive screening and identification of the phytochemicals of VAL and OGL surviving crude oil-contaminated environments like Kokori. The question, therefore, arises whether VAL and OGL produce different and potentially new chemical entities as they adapt to the environmental stress caused by continuous crude oil exploration. Furthermore, not all communities in Delta state are crude oil-rich; an example of a nearby “crude oil-free” community is Abraka [14].

Sickle-cell disease is a multisystem disease accompanied by bouts of acute illness and continuing organ damage restitution. It is one of the world’s most common critical monogenic disorders. The prevalence of the disease has been predominantly reported in resource-restricted countries, and about 80% of SCD incidences eventuate in sub-Saharan Africa. The haemoglobin biophysics and eugenics of the disease have been studied in detail and have helped the understanding of other molecular illnesses. Nevertheless, clinical management of sickle-cell disease is still elementary. Although some evidence confirms support for the utilisation of blood transfusion and hydroxycarbamide, no drugs have been established that explicitly target the pathophysiology of this disease [15].An anomaly characterises sickle cell anaemia resulting from SCD in the haemoglobin (haemoglobin S, HbS) carrying oxygenated blood due to a point mutation substituting a charged and hydrophilic glutamic acid with a hydrophobic valine at position 6 of the β-globin polypeptide chain [16]. Under deoxygenation circumstances (low oxygen tension), the mutant HbS polymerises within the red blood cells (RBCs) into a rigid gel and then into fibres causing a decrease in red blood cells (RBC) deformability. HbS polymerisation occurring within the erythrocytes prompts a modification in the available frame from a typical spheroidal form into disfigured shapes, with some mimicking a sickle [17]. The HbS molecule plays a central role in disease development. Thus, efforts to develop new leads for SCD management should target the inhibition of HbS polymerisation. For example, the recently FDA-approved voxelotor is a covalent modifier of the HbS molecule and acts as an allosteric modulator preserving the haemoglobin molecule in its non-aggregating R-conformer [18].

Normal haemoglobin has a quaternary structure as a tetrameric protein constituent with two α and two β-chains individually and covalently connected to a heme molecule. In the α and β-chains of haemoglobin, there are 141 and 146 amino acids, respectively [19]. Although interactions between several haemoglobin tetrameric units may result in polymerisation, such aggregates are usually unstable under typical physiologic conditions. In homozygous SS patients, the beta-chains contain valine instead of a glutamic acid at position 6, thus enabling it to form hydrophobic connections with identical uncharged hydrophobic units like Phe85 and Leu88 of the adjacent β-globin chain. These connections are secondarily stabilised by other contacts involving segments of the haemoglobin molecule that are remote from the E6V mutation. These contacts form the basis for HbS polymerisation and establish the SCD symptom complex. Normal red blood cells are flexible, can wring through tight capillary junctions and have a lifespan of 90 to 120 days. They circulate across arteries, veins, and tiny capillaries during this period, travelling about 500 km [20]. However, due to the stiffening nature of sickled RBCs, they cannot readily change shape, and the strain of squeezing through tight vascular junctions imparts their tendency to haemolyse; as a result, sickle RBCs have a short lifespan of 10 to 20 days, thus causing anaemia. Only a handful of drugs with significant side effects, including hydroxyurea [21], L-glutamine [22], crizanlizumab [23], and voxelotor [24] are available for managing SCD to date. Hence, there is an urgent need to identify new drugs, particularly plant-derived agents, that are non-toxic and effectively act at various steps of the sickle cell disease pathophysiologic cascade.

The findings in this paper report for the first time the similarities and differences in the antisickling potential and phytochemical profile of *Vernonia amygdalina* (VAL)and *Ocimum gratissimum* (OGL)leaves collected from an explored crude oil-rich Kokori and the crude oil-free species from Abraka in Delta state of Nigeria. To reveal the diversity of the metabolites, untargeted metabolomics analysis was performed to profile diverse classes of the metabolites and compare the overall small-molecule metabolites present in the samples. In order to achieve this, a combination of UPLC separation, QTOF-MS detection, and multivariate statistical evaluation (principal component analysis, PCA) was used to profile the four types of leave samples. Polymerisation inhibition and sickling reversal tests were used as assay methods to evaluate the anti-sickling activities of the different plants’ extracts. In addition, in silico molecular studies were carried out on the compounds isolated from the crude extracts. Our data might provide a compelling basis for further scientific research in Nigeria to investigate the effect of crude-oil exploration-induced environmental stress on the chemical profile, efficacy and safety of commonly consumed medicinal plants in populated rural settings. Findings from such in-depth and elaborate studies could trigger appropriate policy reviews that will mitigate against any possible loss of important plant biodiversity and improve or safeguard the health of the local populations.

## 2. Materials and Methods

### 2.1. Materials and Reagents

VAL and OGL were collected in March 2021 from six different sites; three randomly selected locations within crude oil-rich Kokori and the other three collected from random locations within the crude oil-free Abraka community (Table 1). The plant samples were collected in April 2017 during the dry season, just before the onset of the rainy season. The identities of *Vernonia amygdalina* and *Ocimum gratissimum* leaves were confirmed and authenticated at the Forest Herbarium Ibadan, Nigeria, where voucher specimens were deposited (FHI 10,988 and 109,823). Milli-Q water, methanol and acetonitrile (Rathburn Chemicals Ltd., Walkerburn, UK) were used for the UPLC-MS. Formic acid for UPLC was purchased from Fisher Scientific Ltd., Leicestershire, UK. All chemicals used were of analytical grade. Standard compounds rutin, kaempferol, caffeic acid, catechin, reserpine, eugenol, quercetin, luteolin, isorhamnetin, and oleic acid was purchased from the Nigerian Agency for Food and Drug Administration and Control (Lagos, Nigeria).

### 2.2. Sample Preparation and Extraction

Debris and stalks were removed from the leaves before air-drying, weighed (1.0 g) and extracted in 70% ethanol (1.0 L) for 72 h with occasional stirring. After being filtered, the extraction solution was concentrated using a rotary evaporator (IKA, UK) at 40 °C under reduced pressure. The extracts were further concentrated by freeze-drying and stored at −20 °C for further investigation and analysis.

### 2.3. Mass Spectral Data Acquisition

Extracts were dissolved in methanol at a final concentration of 0.1 mg/mL, centrifuged and injected onto a Phenomenex Kinetex XB-C18 (2.6 mM, 100 × 2.1 mm) column. Samples were analysed using a Bruker maXisII electrospray ionisation quadrupole time-of-flight (ESI-qToF) mass spectrometer coupled with an Agilent 1290 UHPLC system. The elution conditions were 5% ACN + 0.1% FA to 100% ACN + 0.1% FA in 15 min. Mass range was set to *m*/*z* 100–2000. The positive and negative mode conditions were as follows: capillary voltage 4.5 kV, nebuliser gas 4.5 bar, dry gas 12.0 L/min, and dry temperature of 250 °C. MS/MS experiments were conducted under Auto MS/MS scan mode with no step collision. The external reference lock mass (sodium formate) was infused at a constant flow of 0.1 mL/h.

### 2.4. Phytochemical Analysis of Metabolites of VAL and OGL Using UNIFI Platform

The UNIFI 1.7.0 software (Waters, Manchester, UK) [25], Ref. [26] was first used to identify the chemical compounds present in all the plant samples; the MS raw data (.d) was automatedly screened using an established streamlined workflow. The parameters used were fine tunned as follows: minimum peak area was set to 200 for 2D peak detection, peak intensities for low and high energy were set above 1000 and 200 counts consecutively for 3D peak detection; 5 ppm for mass error, and 0.1 min retention time to match the standard reference. The speculated fragments were recognised as the matching compounds. Both negative and positive adducts were selected in the analysis. Reserpine was used as the reference compound, [M + H]^+^ 609.6773 was used as the positive ion and [M − H]^−^ 607.6617 was the negative ion. Phytochemicals were further verified by comparing generated fragmentation patterns with characteristic MS fragmentation patterns in published literature and comparing reference compounds with retention time. The investigation of the phytochemicals was carried out systematically. In addition to the in-built Waters UNIFI Traditional Medicine Library, the following databases were used to source chemical information for the compounds: ChemSpider, Pubmed, Pubchem, Reaxys, and Web of Science. These aided the formation of a self-built database of phytochemicals isolated from VAL and OGL.

### 2.5. Metabolomic Analysis

The crude extracts’ obtained HPLC-MS/MS data were converted from data analysis (.d) to .mzXML file format using MSConvert software [27]. The .mzXML files were further processed using MZmine software [28]. The following parameter modules were used: mass detection (RT 2.5–35 min, centroid); chromatogram builder (MS level 1; minimum height 1.0 × 104; minimum time 0.5 min; *m*/*z* tolerance 10 ppm); deconvolution of the spectra (Algorithm Savitzky-Golay); isotopic peaks grouper (*m*/*z* tolerance 10 ppm; retention time (RT) tolerance 0.1 min); duplicate peak filtering; smoothing; data alignment (Join aligner; *m*/*z* tolerance 10 ppm; RT tolerance 0.5 min); gap-filling (intensity tolerance 20%; *m*/*z* tolerance 10 ppm, RT tolerance 0.5 min); and peak filtering range (0.00–0.5 min). After processing the data, peaks with matching RT and *m*/*z* in different samples were regarded as belonging to the same component. Furthermore, multivariate statistical analysis was performed. Firstly, principal component analysis (PCA) showed pattern recognition and maximum variation to obtain the overview, classification and variables that emerge as significant to get the maximum separation between the crude oil-contaminated VAL and OGL and the crude oil-free VAL and OGL groups and explore the potential phytochemicals that contribute to the differences. Then, S-plots were created to provide visualisation of the OPLS-DA to enable model elucidation. Simultaneously, the use of variable importance for the projection (VIP) helped screen the different phytochemicals, and compounds with VIP values >1.0 and *p*-value below 0.05 were considered significant [26]. Additionally, permutation testing was performed to indicate the R2/Q2 values that could indicate statistical significance [29]. Finally, Simca 15.0 software (Umetrics, Malmö, Sweden) was used to show the analysis results [30].

### 2.6. Blood Sample Collection and Ethical Approval

Blood samples from sickle cell anaemia (HbSS) patients were collected from the University College Hospital, Ibadan, Nigeria. 4 mL of blood samples were collected by venipuncture directly into ethylene diamine tetra acetic acid (EDTA) anticoagulant bottles. The blood samples were centrifuged and washed with phosphate-buffered saline pH: 7.4 (PBS: 1.3 M NaCl, 0.07 MNa_2_HPO_4_ and 0.03 M NaH_2_PO_4_) at 3000 rpm for 5 min. All experimental protocols were conducted and compliant with the ethical approval of the University of Ilorin Ethics Committee Guidelines and internationally accepted principles as found in US guidelines (NIH publication #85-23, revised in 1985).

### 2.7. In Vitro Sickling Reversal Assay

A sickling reversal test was carried out using the method previously described in the literature [31]. 1 mL of HbS blood collected was added to 5 mL of phosphate-buffered saline (PBS- pH 7.4) in a centrifuge tube. The content was centrifuged at 3000 rpm for 15 min. The supernatant was discarded, and the process was repeated to obtain a clear supernatant. The red blood cells were re-suspended with 1 mL of PBS and used for the test. 40 μL of the washed red blood cells were pipetted into a clean Eppendorf tube using a micropipette. 30 μL of PBS was added to the Eppendorf tube containing the washed blood. 800 μL of freshly prepared 2% sodium metabisulfite was added and incubated in a thermostated water bath at 37 °C for 1 h. 200 μL of PBS was added after the initial incubation period and incubated for another 1 h at 37 °C. 10 μL of the incubated cells were transferred onto a hemocytometer. The cells were counted at five zones using a differential counter (control); this process was repeated by substituting the 200 μL of distilled water added after the first incubation with 200 μL each of the extracts at three different concentrations (10, 5 and 2 mg/mL). The cells were classified as normal or sickle using visual inspection of their shapes.

Calculation:

Normal cells = Biconcave or disk-like shapes

Sickle cells = star-like or wrinkled shapes
$$\%\;\mathrm{Sickled}=\frac{\mathrm{Number}\;\mathrm{of}\;\mathrm{sickle}\;\mathrm{cells}}{\mathrm{Total}\;\mathrm{number}\;\mathrm{of}\;\mathrm{cells}}\times 100$$
$$\%\;\mathrm{Reversal}=\frac{\left(\%\;\mathrm{sickled}\;\mathrm{cells}\;\mathrm{control}-\%\;\mathrm{sickled}\;\mathrm{cells}\;\mathrm{test}\right)}{\%\;\mathrm{sickled}\;\mathrm{cells}\;\mathrm{control}}\times 100$$

### 2.8. Isolation of Secondary Metabolites

1D and 2D NMR data were recorded on Bruker AVANCE II 600 MHz and 400 MHz spectrometers. The spectra were obtained at 300 K in CD_3_OD. The CD_3_OD solvent signals (^1^H; 3.31 ppm and ^13^C; 49.1 ppm) were used to reference the spectra. HR-ESI-MS data were obtained using an Agilent 6540 HR-ESI-TOF-MS coupled to an Agilent 1200 HPLC system. Fractionations were performed on a solid-phase extraction column using C18-E (Phenomenex, 55 μm, 70 Å, 2 g/12 mL, Giga tubes). Fractions were further purified on an Agilent 1200 semipreparative HPLC system equipped with a binary pump, photodiode array detector (DAD)22, Waters Sunfire reversed-phase column C18 (5 μm, 10 × 250 mm), and Agilent Zorbax C18 (5 μm, 9.4 × 250 mm), and a mobile phase gradient system between 95:5% and 20:80% (H_2_O/MeOH).

### 2.9. Molecular Docking

The crystal structure of deoxyhemoglobin S (PDB ID: 2HBS) [32] and P-selectin (PDB ID: 1G1S) [33] was obtained from the protein data bank (https://www.rcsb.org/ , accessed: 28 September 2022). Prior to protein preparation, heteroatoms and co-crystallised ligand were removed. The dock prep module of the UCSF chimera software program (University of California, Oakland, CA, USA) [34] was used for protein preparation. Briefly, solvent molecules co-crystallised with the proteins were deleted; hydrogen atoms were added to the protein. In addition, the Dunbrack 2010 rotamer library [35] was used to predict the missing side chains, and charges were defined using AM1-BCC. The protein structure was minimised with the AMBERff14SB force field. The minimisation parameters include 300 steepest descent steps at a step size of 0.02 Å, conjugate gradient steps at 10 and an update interval of 10. Subsequently, the protein was loaded on PYRX for molecular docking [36]. The ligand; vernodalol was downloaded from pubchem.ncbi.nlm.nih.gov/ accessed on 28 September 2022) with Pubchem ID 442318. The structure of lasiopulide was identified from [37] and constructed using ChemDraw (RRID: SCR_016768). The ligands were imported to PYRX, and the docking parameters are shown in the Table 2:

### 2.10. 3D Pharmacophore Model Generation

LigandScout by Inte Ligand, Expert software (Wolber and Langer), Vienna, Austria, Europe [38] (license key: 44427459425915253797) was used to generate a 3D pharmacophore model. Espresso algorithm was used to generate ligand-based pharmacophore. The generated pharmacophore model compatibility with the pharmacophore hypothesis was created using default settings for LigandScout. Relative Pharmacophore-Fit scoring function, Merged feature pharmacophore type and feature tolerance scale factor was set to 1.0 for Ligand-Based Pharmacophore creation. The best model was selected from the 10 generated models.

### 2.11. Statistical Analysis

The results obtained from all experiments were performed in triplicates and expressed as the mean ± SEM. The statistical significance of the means differences was established by analysis of variance (ANOVA) with Duncan’s post hoc tests. *p*-values < 0.05 were considered statistically significant.

## 3. Results and Discussion

### 3.1. Identification of Phytochemicals from VAL and OGL Based on the UNIFI Platform

The screening analysis revealed that 70 compounds were identified or tentatively characterised in ESI^+^ and ESI^−^ mode from VAL and OGL from crude oil-rich Kokori and crude oil-free Araka. Seven shared constituents were identified in VAL and OGL from both communities (VAL-AB, OGL-AB, VAL-KK and OGL-KK); 6 constituents were exclusively identified in VAL-KK and OGL-KK. More specifically, 34 compounds were characterised from VAL-AB, 20 from VAL-KK, 39 from OGL-AB and 21 from OGL-KK. (Appendix A). Samples obtained from crude oil-free Abraka (VAL-AB and OGL-AB) displayed richer chemical profiles with various structural patterns, including flavonoids, glycosides, organic acids, iridoids, saponins and steroids, compared to samples collected from crude oil-spill Kokori (VAL-KK and OGL-KK). The base peak chromatograms marked with the number of compounds are shown in Appendix A. The structures of the phytochemicals are summarised in Figure 1.

From the screening analysis, 34 compounds were characterised from VAL-AB, 21 from VAL-KK, 39 from OGL-AB and 21 from OGL-KK. Altogether 60 compounds were identified in positive mode, and 10 were identified in both negative and positive modes. From the BPI chromatograms (Appendix A), the positive ionisation mode was better than the negative mode based on the number of compounds ionised and the fragmentation patterns of the identified compounds. The results also showed that both VAL and OGL from Abraka (crude oil-free community) are richer in phytochemicals. These plants have previously reported rich, diverse metabolites [39]. The main chemical compositions identified in this study were various terpenes, flavonoids, and steroids. Interestingly, a handful of compounds unique to VAL-KK and OGL-KK samples collected from Kokori, the crude oil-spill community, were also identified. These compounds are steroids (38, 42) and terpenes (33, 47, 57 and 69). This comprehensive phytochemical profile study revealed the structural diversity of secondary metabolites and matching patterns within VAL and OGL; some similarity was established between VAL and OGL regarding the synthesised compounds. Seven compounds were shared constituents found in all VAL and OGL samples used for this study, irrespective of their collection sites. These are dihydroquercetin (7), caffeic acid (10), linolenic acid (19), isoquercetin (34), rutin (36), *tetranotreterpene* (59) and oleic acid (70). Furthermore, in the nontargeted metabolomic analysis performed and based on the secondary metabolites identified, it was found that there indeed existed differences between VAL-AB and VAL-KK, OGL-AB and OGL-KK, respectively.

For VAL, 13 known metabolites, including eight steroids (31, 33, 38, 42, 44, 53, 57 and 58), two organic acids and organic acid ester (9), one flavonoid (56), two terpenes (47 and 69) and one unknown compound (55) enabling this differentiation were discovered. These compounds were identified only in VAL samples from Kokori, the crude-oil spill community, thereby illustrating the differences between VAL-AB and VAL-KK and, therefore, providing a basis for explaining the effect of different growth environments on secondary sources metabolites. Among these potential biomarkers, compounds 31, 38, 42, 44, 47, 53, 56, and 57 were identified or tentatively characterised in *Vernonia amygdalina* leaves for the first time. As previously stated, a total of 20 compounds were identified in VAL-KK, 13 of which were exclusive to VAL-KK, while the remaining 7 (7, 10, 17, 19, 34, 36 and 70) compounds were identified in both VAL-AB and VAL-KK samples.

With *Ocimum gratissimum* leaves, nine unique compounds (25, 33, 38, 42, 47, 57, 61, 62 and 69) were identified from OGL-KK. These phytochemicals have never been reported in the literature identified or isolated from OGL. Interestingly, some similarity was displayed between samples collected from Kokori, the crude oil-spill community. These metabolites were identified in VAL-KK and OGL-KK; 33, 38, 42, 47, 57 and 69.

### 3.2. Diversity Evaluation of VAL and OGL Using Metabolomic Analysis

The quality control (QC) samples were clustered tightly within the same quadrant in PCA, suggesting the system’s reliability, reproducibility, and stability. Based on their common spectral characteristics, the PCA 2D plots of VAL and OGL samples from crude oil-spill Kokori and crude oil-free Abraka were easily classified within two clusters (Figure 2); they were separated, indicating that samples from Kokori could be easily differentiated from samples from Abraka. The first two components, PC1 and PC2, explained 78.2% and 16.5% of the total variance in the positive mode and 68.2% and 17.2% in the negative mode.

In order to evaluate the variance between the plant samples from the crude oil-spill community (Kokori) and the crude oil-free community (Abraka), the OPLS-DA score plot, permutation test, and variable importance in the projection values were obtained. After OPLS-DA plots (Figure 3a and Figure 4a) in both ESI^+^ and ESI^−^ modes were performed, the maximum separation between the VAL, OGL and, VAK, OGK groups were available.

With the permutation testing done, the Q2 and R2 values obtained were 0.823 0.721 which suggests that the model can be considered predictive and reproducible. The lines grouping the crude oil-spill and crude oil-free samples were significantly located underneath the random sampling lines (Figure 3b and Figure 4b), indicating a definite validity for the characteristic metabolites identified. Additionally, a heatmap was generated to systematically evaluate the richness of the secondary metabolomes in all the samples (Figure 5); this visualised the metabolite’s intensities and diversity in the extracts. Samples were run in triplicates. Multiple blue bands indicate a rich secondary metabolome with a high diversity of metabolites, while fewer blue bands indicate a more limited set of secondary metabolites. The heat map was also arranged according to the collection sites to investigate the chemical diversity among plant samples from the same species. OGL-AB displayed the richest metabolome of all four extracts.

### 3.3. In Vitro Antisickling Activities of Crude Extracts

Reduction of Sickle cells and Sickling Reversal assay.

The inhibition and reversal of sickling by hydroxyurea (positive standard) and plant extracts (test samples) were conducted to investigate and compare the effect of Vernonia amygdalina and Ocimum gratissimum extracts collected from the crude oil-rich Kokori and crude oil-free Abraka, Delta State, Nigeria on sickle cell disease. The two study medicinal plants are used within the tribal communities of Abraka and Kokori in Delta state, Nigeria, to manage blood-related diseases such as sickle cell anaemia. The percentage of sickling inhibition and reversal were used in vitro to determine the possible anti-sickling properties of the two plant species on HbSS erythrocytes. Sickle cell disease negatively affects the flexibility of the red blood cell and thus disallows them from their natural kinetic activity with the blood vessels. Antisickiling drugs are therefore designed to intrinsically or otherwise prevent or reverse the sickle shape appearance of the red blood cells of HBSS patients. Data obtained from the sickling inhibition experiment and the sickling reversal assay have been presented in Table 3, Figure 6 The effect of the plant extracts was compared with a standard foremost drug, hydroxyurea. Data obtained from this study indicates that the extracts of the two plants possess sickling inhibition (OGL-AB = 28.43%; VAL-AB = 26.89%; OGL-KK = 26.17%; VAL-KK = 26.29%, Hy.urea = 26.36%) (Figure 6) and sickling reversal activities (OGL-AB = 35.31%; VAL-AB = 33.52%; OGL-KK = 29.73; VAL-KK = 30.22%; Hy.urea = 34.85%) comparable to or marginally higher than the standard drug, hydroxyurea (Table 3). However, in both sickling inhibition and reversal assays, VAL and OGL extract collected from crude oil-free Abraka displayed higher activities than VAL and OGL extracts originating from crude oil spill Kokori; this may be due to the reduction in the number of phytochemicals in the samples from Kokori. As seen in the metabolomic studies (Appendix A, Figure 2 and Figure 3), samples collected from crude oil-free Abraka showed a more diverse and richer phytochemical profile than samples from Kokori; this is in agreement with published literature which shows that natural products with richer metabolomes tend to display higher bioactivities [40].

The number of cells identified as sickled in the control was over 40%, while in the presence of hydroxyurea, sickling was reduced to 26.36%. Generally, all four extracts tested produced a slight reduction in the number of sickling red blood cells. Ocimum gratissimum from Abraka displayed the highest sickling inhibition of 28.43% and sickling reversal of 35.31%. It has previously been reported for its activities against sickling of the HbS [41], while Tshilandi et al. (2015) [2] isolated the bioactive component of the plant and documented ursolic acid as the antisickling agent. Findings from this study are consistent with these reports regarding the antisickling potentials of Ocimum gratissimum. Vernonia amygdalina has only been reported once [42] for its potentiating activities of polymerisation of red blood cells.

### 3.4. Structural Elucidation of Compounds Isolated

The methanolic plant extracts, VAL-AB and VAL-KK, were selected for structural elucidation due to their large masses and analysed using HPLC and ^1^H NMR. The ^1^H NMR showed a wide range of interesting signals that indicated the extract’s chemical diversity. The VAL-AB and VAL-KK extracts were fractionated and purified according to the materials and methods. Compound 1, vernodalol [43] was isolated from VAL-AB and VAL-KK, while compound 2, lasiopulide [44] was isolated exclusively from VAL-KK. Both compounds were identified by comparing their experimental NMR and HRESIMS *m*/*z* data with published literature (Appendix A).

### 3.5. Molecular Docking Studies

In recent study, molecular modelling methods have been useful in evaluating the interaction between target molecules and ligands [45]. After identifying the antisickling activities of the crude extracts, the isolated compounds were subjected to virtual screening by molecular docking to predict the interactions between selected target proteins and the molecule and elucidate the possible mechanism of action. Two protein targets, deoxyhemoglobin S (PDB ID: 2HBS) and P-selectin (PDB ID: 1G1S) were obtained from the protein data bank and simulated with vernodalol (compound 1) and lasiopulide (compound 2)**.** The ligands, Protoporphyrin IX and (4R)- 2-Methylpentane-2,4-diol are co-crystallised ligands of deoxyhemoglobin S (PDB ID: 2HBS) and P-selectin (PDB ID: 1G1S), respectively. 4-{2-chloro-4-[3-(1H-imidazol-2-yl)propanoyl]phenoxy}butanoic acid and furfural were also selected for the docking studies, having been previously described to possess antisickling effect via deoxyhaemoglobin and haemoglobin [46,47].

P-selectin is a cellular adhesion receptor that manifests on endothelium and activated platelets and facilitates vaso-occlusion in sickle cell disease (SCD). It modulates the interactions between leukocytes and activated platelets; and participates in the capturing, rolling and recruiting of leukocytes to the activated vascular endothelium. The interaction with platelets triggers the recruitment of monocytes and neutrophils, which promote inflammation [48]. The inhibition of P-selectin has been shown to limit the frequency of vaso-occlusive crises in patients with SCD. Additionally, P-selectin–mutant showed poor leukocyte rolling and retards the leakage of neutrophils to sites of inflammation [49]. An inhibitor that targets P-selectin reduces the pain duration that accompanies sickle cell crises with a significant reduction in opioid use [50]. Lasiopulide and vernodalol demonstrated a better affinity for the P-selectin receptor (PDB ID: 1G1S) with a binding energy of −20.9200 and −20.7568 kJ/mol, respectively than the co-crystallised ligand ((4R)-2-Methylpentane-2,4-diol) (−15.0624 kJ/mol); this shows the prospect that these phyto-compounds could help in reducing vaso-occlusion in sickle cell disease (SCD) (Figure 7 and Figure 8).

It has been reported that small molecules with a high affinity for HBS or directly modified protein could hinder the polymerisation of this molecule which is the root cause of sickle cell pathology [47]. Agents such as hydroxyurea (HU) currently used in managing and treating SCD may trigger some adverse reactions like myelosuppression. Thus, the search for new antisickling agents is inevitable. Lasiopulide and vernodalol displayed an affinity for HBS than furfural; a previously reported compound with affinity for HBS [46]; but less than the other co-crystallised ligands (protoporphyrin IX (HEM) and 4-{2-chloro-4-[3-(1H-imidazol-2-yl) propanoyl] phenoxy}butanoic acid (3U8) [47] as shown in Table 4.

The docked structure, vernodalol (Figure 7), exhibited interaction with the active site of P-selectin via TYR3, HIS4, TYR5, and VAL41 residues. This compound demonstrated two H-bond interactions with TYR3, a conventional hydrogen bond and a carbon hydrogen bond, and a pi-alkyl interaction with His4 and TYR5. Docking analysis revealed a Pi-alkyl interaction between lasiopulide and the TYR5 residue and a two-carbon hydrogen bond with the P-Selectin GLY138 residue. According to the results of Vernodalol docking studies with deoxyhemoglobin, a carbon-hydrogen bond exists between the ligand and histidine residue at positions 58 and 87 of the target, while the Pi-alkyl interaction was found with LEU101, VAL62, and PHE43. Lasiopulide with this same target displayed a conventional hydrogen bond with PRO95 and VAL96 and a pi-alkyl interaction with PHE98. P-interactions have been reported to contribute to stabilising binding structures, and the frequency of hydrogen bonds is proportional to strong inhibitor binding. Alkyl and pi-alkyl bonds have also been reported to improve the hydrophobic interaction of the ligand in the binding pocket of the protein [51].

### 3.6. Pharmacophore Evaluation

Using the lowest energy conformers of vernodalol and lasipulide, the pharmacophore models were generated [46]. The generated pharmacophore showed three key features: hydrogen bond acceptors (HBAs), hydrogen bond donors (HBDs), and hydrophobic interactions (H). The representative 3D and 2D pharmacophoric features of each compound are shown in Figure 9 and Figure 10. Each compound constitutes individual pharmacophoric features and from these individual characteristic pharmacophores were generated. A merged pharmacophore of vernodalol with common features was generated, as shown in Figure 9. This common feature pharmacophore model of vernodalol with a score of 0.9909 showed certain features: two HBD, five HBAs, and three Hs.

Another merged pharmacophore of lasiopulide with common features was generated, as shown in Figure 10. This common feature pharmacophore model of lasipulide with a score of 0.9889 showed certain features: two HBD, four HBAs, and two Hs.

## 4. Conclusions

In this study, the similarity and diversity of phytochemical profile of *Vernonia amygdalina* and *Ocimum gratissimum* samples collected from crude oil-free community (Abraka) and the crude oil-spill community (Kokori) were evaluated. The crude oil-spill community displayed a more prosperous and diverse phytochemical profile than the same samples originating from the crude oil-spill community. The presence of secondary metabolites such as lasiopulide found exclusively in VAL-KK and OGL-KK strongly supports the hypothesis that the environmental condition induced by crude oil spillage, thus affect the phytochemicals biosynthesis of the plants [52]. Therefore, this may support our assumption that medicinal plants surviving unfavourable crude oil spills may produce potentially new compounds as they adapt to environmental stress.

From the molecular docking studies, vernodalol and lasiopulide displayed an affinity for P-selectin and deoxyhemoglobin S, which are proteins implicated in sickle cell disease; thus, they warrant further investigations such as in vivo and mechanism of action studies. There appears to be a slight difference between the antisickling activities of OGL AND VAL growing in the crude oil-rich Kokori and that growing in crude oil-free Abraka; The impact of the different collection sites on the bioactivities and phytochemical profiles was highlighted by the corresponding differences in chemical profiles and the invitro ant sickling assay. Pharmacophore models were proposed to help guide future drug design studies. Additionally, the proposed pharmacophore models should be used as a future guide for selecting and designing antiparasitic drug candidates. Further studies is therefore warranted to identify and characterise the active phytochemicals present in different active extracts, followed by toxicity analysis and clinical trials.

## Figures and Tables

**Figure 1 molecules-27-08372-f001:**
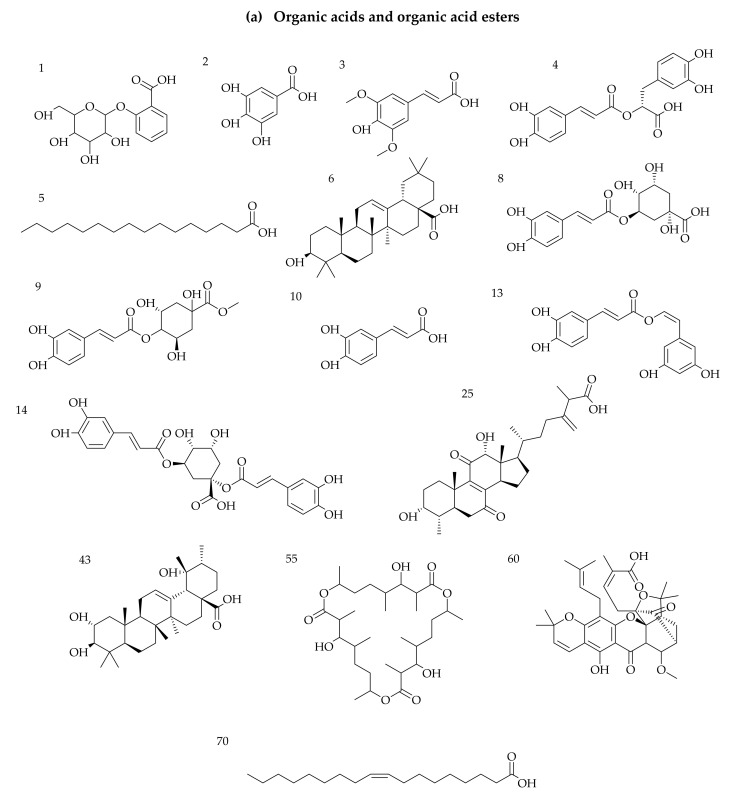
(**a**–**d**) Chemical structures of compounds identified in VAL and OGL from Kokori and Abraka. (**a**) organic acids and organic acid esters, (**b**) flavonoid, (**c**) terpenes (**d**) Others.

**Figure 2 molecules-27-08372-f002:**
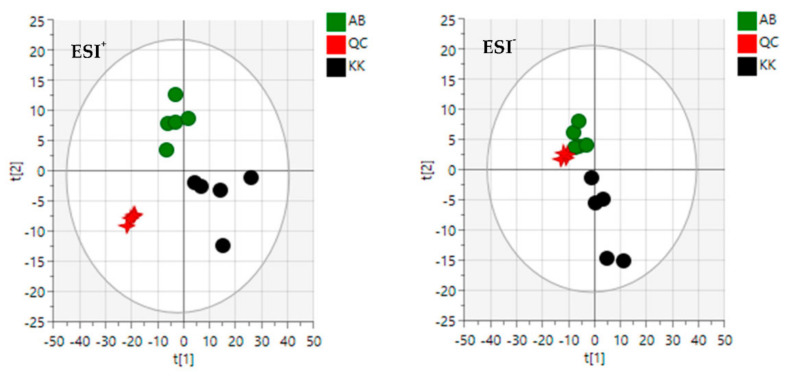
The principal component analysis (PCA) of VAL and OGL in ESI^+^ mode and ESI^−^ mode. AB—Abraka; KK—Kokori; QC—Quality control.

**Figure 3 molecules-27-08372-f003:**
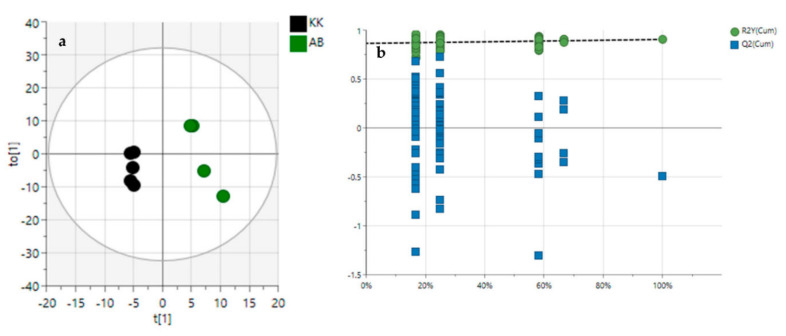
Orthogonal partial least squares discriminant analysis (OPLS-DA) (**a**), permutation tests, (**b**) in ESI^+^ mode. AB—Abraka; KK—Kokori.

**Figure 4 molecules-27-08372-f004:**
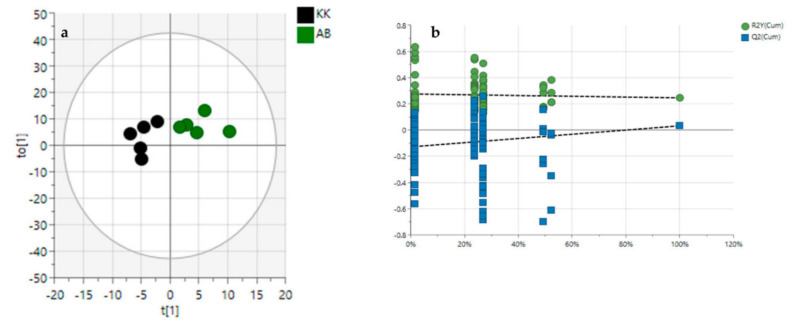
Orthogonal partial least squares discriminant analysis (OPLS-DA) (**a**), permutation tests, (**b**) in ESI^−^ mode. AB—Abraka; KK—Kokori.

**Figure 5 molecules-27-08372-f005:**
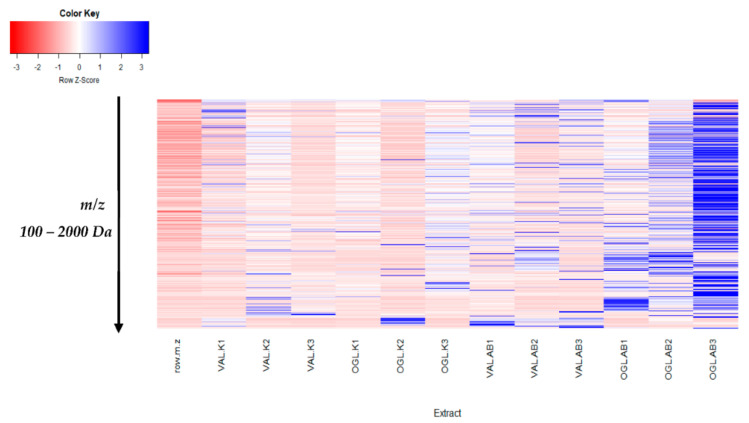
Heatmap visualising the intensities of secondary metabolites. AB-Abraka; KK—Kokori; VAL—*Vernonia amygdalina* leaves; OGL—*Ocimum gratissimum* leaves.

**Figure 6 molecules-27-08372-f006:**
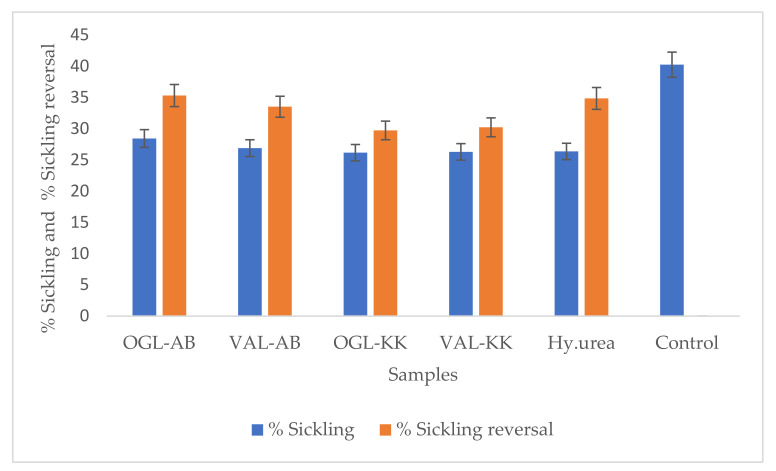
Percentage sickling reduction and sickling reversal of erythrocytes treated with 10 mg/mL extracts. VAL—Vernonia amygdalina leaves from Kokori; OGL-AB—Ocimum gratissimum leaves from Abraka; Hy.urea—Hydroxyurea.

**Figure 7 molecules-27-08372-f007:**
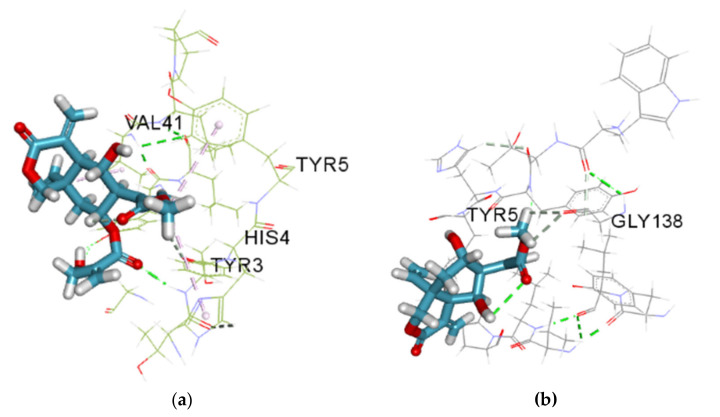
(**a**) Vernodalol in the active site of P-Selectin (PDB ID: 1G1S). Predicted hydrogen bonding interactions can be seen between vernodalol and active residues VAL 41, TYR5, TYR3 and HIS4. (**b**) Interaction between lasiopulide and P-selectin (PDB ID: 1G1S). Conventional Hydrogen bond (solid green colour), carbon-hydrogen bond (solid grey colour) and Pi-Alkyl (solid lilac colour).

**Figure 8 molecules-27-08372-f008:**
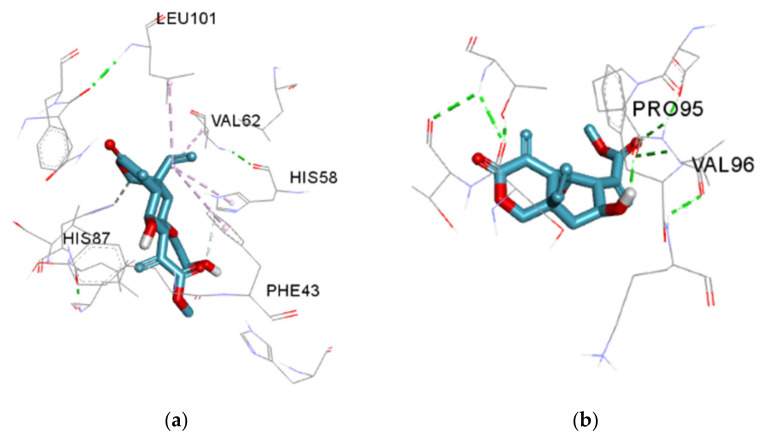
Deoxyhemoglobin S (PDB ID: 2HBS) (**a**) Lowest energy docked pose of vernodalol (**b**) Lowest energy docked pose of lasiopulide. Conventional Hydrogen bond (solid green colour), carbon-hydrogen bond (solid grey colour) and Pi-Alkyl (solid lilac colour).

**Figure 9 molecules-27-08372-f009:**
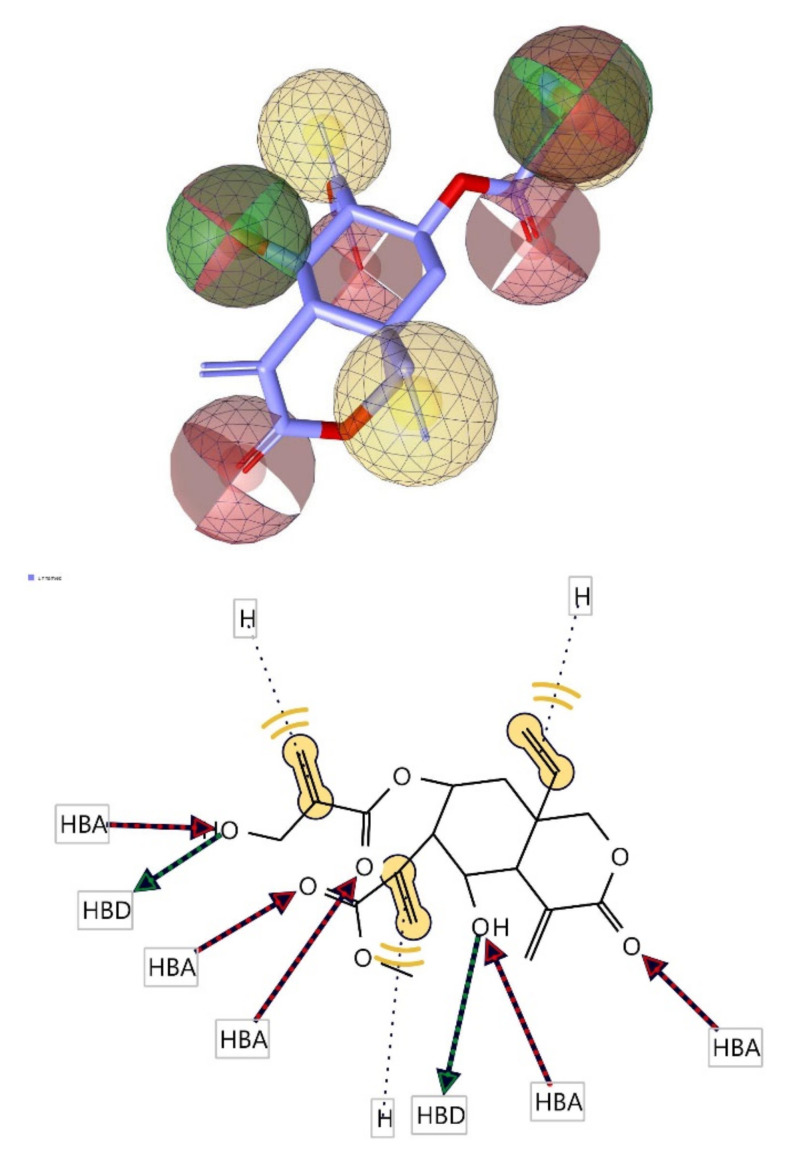
3D and 2D representations of common pharmacophoric features of vernodalol used in pharmacophore evaluation. Red, HBAs; green, HBDs; Yellow, H; as described earlier.

**Figure 10 molecules-27-08372-f010:**
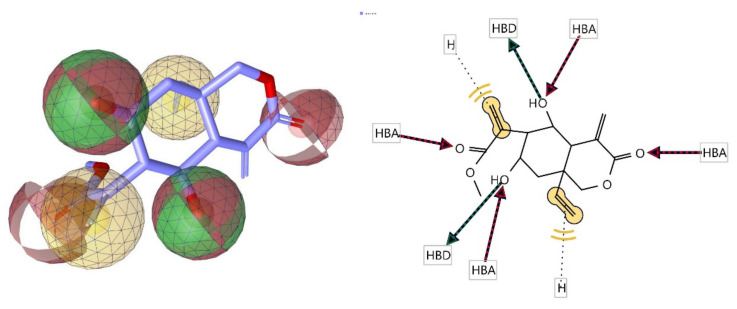
3D and 2D representations of common pharmacophoric features of lasiopulide used in pharmacophore evaluation. Red, HBAs; green, HBDs; Yellow, H; as described earlier.

**Table 1 molecules-27-08372-t001:** List of tested samples from Kokori and Abraka, Delta State, Nigeria. Legend: VAL, *Vernonia amygdalina* leaf; OGL, *Ocimum gratissimum* leaf.

^#^ Species	Sample No.	Community	Source	Coordinates
VAL	1	Abraka	Oria	5°45′ N 6°15′ E
	2	Abraka	Urhuovie	5°47′ N 6°15′ E
	3	Abraka	Ajalom	5°47′ N 6°15′ E
	4	Kokori	Egbo	5°38′ N 6°04′ E
	5	Kokori	Kokori	5°38′ N 6°15′ E
	6	Kokori	Samagidi	5°38′ N 6°22′ E
OGL	7	Abraka	Oria	5°45′ N 6°15′ E
	8	Abraka	Urhuovie	5°47′ N 6°15′ E
	9	Abraka	Ajalom	5°47′ N 6°15′ E
	10	Kokori	Egbo	5°38′ N 6°04′ E
	11	Kokori	Kokori	5°38′ N 6°15′ E
	12	Kokori	Samagidi	5°38′ N 6°22′ E

^#^ Plant samples were collected in April 2017, towards the end of the dry season.

**Table 2 molecules-27-08372-t002:** Docking parameters.

Target Name	PDB Code	Dimensions (Å)	Centre
deoxyhemoglobin S	2HBS	X: 24.4838; Y: 26.6805; Z: 25.0000	X: 7.5857; Y: 11.1820; Z: 28.1205
P-selectin	1G1S	X: 34.1555; Y: 21.0892; Z: 25.0000	X: 43.7339; Y: 85.8462 Z: 48.1365

**Table 3 molecules-27-08372-t003:** Percentage of sickling of erythrocytes treated without extracts (negative control), hydroxyurea (positive control) and with 10 mg/mL extracts.

Samples (Extracts)	% Sickling ± SD	% Sickling Reversal ± SD
OGL-AB	28.43 ± 0.50	35.31 ± 0.30
VAL-AB	26.89 ± 0.51	33.52 ± 0.30
OGL-KK	26.17 ± 0.16	29.73 ± 0.12
VAL-KK	26.29 ± 0.46	30.22 ± 0.50
Hy.urea	26.36 ± 0.02	34.85 ± 0.16
Control	40.25 ± 0.32	0.00

VAL—Vernonia amygdalina leaves from Kokori; OGL-AB—Ocimum gratissimum leaves from Abraka; Hy.urea—Hydroxyurea.

**Table 4 molecules-27-08372-t004:** Docking energies (KJ/mol) of the isolated compounds with target proteins.

Compounds	Binding Affinity (KJ/mol)	
	1G1S	2HBS
Lasiopulide	−20.9200	−23.4304
Vernodalol	−21.7568	−22.5936
(4R)-2-Methylpentane-2,4-diol	−15.0624	−17.9912
Furfural	−13.8072	−16.736
4-{2-chloro-4-[3-(1H-imidazol-2-yl)propanoyl]phenoxy}butanoic acid	NB	−31.7984
Protoporphyrin IX	NB	−43.932

Receptors and PDB ID. 1G1S: P-Selectin; 2HBS: Deoxyhemoglobin S; NB: Non-binding.

## Data Availability

Not applicable.

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
