# Peer review of "Can Crude Oil Exploration Influence the Phytochemicals and Bioactivity of Medicinal Plants? A Case of Nigerian Vernonia amygdalina and Ocimum gratissimum"

_molecules, 2022, doi:10.3390/molecules27238372_

Round 1

Reviewer 1 Report

Diyaolu et al. Studied the effect of crude oil on the phytochemicals of Medicinal plants (Vernonia amygdalina and Ocimum gratissimum) and also compared their results with oil-free zone plants. The authors identified several novel compounds using in vitro and in silico methods. As expected plants from crude oil-free regions were found rich in phytochemicals and have better anti-sickling effects. Overall, this is a well-presented work and can be published after minor corrections. Here are a few suggestions for authors:

Please rephrase line number 21-22 in the Abstract.

Please revise the introduction by removing the unnecessary data for example paragraphs from Line number 112 and 127 can be reduced into one.

It would be great if the authors discuss a few potential compounds currently in use against SCD.

Instead of Table3, Figure S3 can be more informative in the manuscript. Therefore, the authors can move Table S3 to a supplementary file.

In molecular docking results Table5 how the native ligand of PDB 1G1S does not display any affinity against the target. Moreover, both the selected compounds displayed 50% less binding affinity towards the target 2HBS than the co-crystal ligand. Does this is correct to say that the selected protein is the potential target for these compounds?

Please improve Figure 10. It is difficult to see hydrogen bonds. Non-polar hydrogens can be removed from ligands.

Please revise the conclusion section to key findings.

Please uniform special characters such as in silico or in silico

Author Response

  1. Please rephrase line number 21-22 in the Abstract

Authors’ response: The sentence has been rephrased in the revised version of the manuscript(line number 21-23). Thank you

  1. Please revise the introduction by removing the unnecessary data for example paragraphs from Line number 112 and 127 can be reduced into one

Authors’ response: This section of the introduction has been revised (please see line number 39-118)

  1. It would be great if the authors discuss a few potential compounds currently in use against SCD.?

Authors’ response: This information has been added to line number 89-92 and 109-111.

  1. Instead of Table3, Figure S3 can be more informative in the manuscript. Therefore, the authors can move Table S3 to a supplementary file

Authors’ response: We have replaced Table 3 with Figure 3 as suggested (please see line numbers 303-334).

  1. In molecular docking results Table 5 how the native ligand of PDB 1G1S does not display any affinity against the target. Moreover, both the selected compounds displayed 50% less binding affinity towards the target 2HBS than the co-crystal ligand.

Authors’ response:

Clearer sentences were included in the section to explain the different ligands used in the doking analysis. 4-{2-chloro-4-[3-(1H-imidazol-2-yl)propanoyl]phenoxy}butanoic acid and Protoporphyrin IX are not native ligands for P-Selectin (PDB: 1G1S). Rather, they are ligands for deoxyhemoglobin, while (4R)-2-Methylpentane-2,4-diol is the ligand co-crystal( please see line numbers 541-545).  The docking result shows that Lasiopulide and Vernodalol do have affinity for deoxyhemoglobin, though less than that of Protoporphyrin IX which is a constituent of hemoproteins.

Does this is correct to say that the selected protein is the potential target for these compounds?

Authors’ response: Yes. The aim of the docking study is to showcase the likely potential target of the natural compounds against the P- selectin and haemoproteins.

  1. Please improve Figure 10. It is difficult to see hydrogen bonds. Non-polar hydrogens can be removed from ligands.

Authors’ response: Figure 9 and 10, now shown as Figures 7 and 8 have been improved in the revised manuscript( Please check line numbers 641 and 670)

  1. Please revise the conclusion section to key findings.

Authors’ response: The authors have carefully revised the key findings in the conclusion (line number 650-670)

  1. Biological data. The authors now state in the experimental that measurements were recorded in triplicate, however only 2 data points are visible for each compound concentration, and only one for the controls. This must be clearly explained, along with a description of the data analysis performed.

Authors’ response: Authors thank the reviewer for this comment. For clarity, Table 3 has now been simplified and harmonised. Since three determinations/readings were taken, the standard deviations have been added to the table. In addition, in Figure 8, the graph has equally been harmonised and simplified to show the antisickling responses as well as inhibitory responses.

  1. Please uniform special characters such as in silico or in silico

Authors’ response: It has been corrected in the revised manuscript. Thank you.

Reviewer 2 Report

This is an outstanding paper that provides a very detailed analysis of  the extracts from  two  Nigerian medicinal plants, Vernonia amygdalina  (VAL) and Ocvimum gatissimum (OGL) that are able to survive not only drought and harsh conditions but also  crude oil spillage. The abstract  summarizes the  detailed work well. The introduction provides a detailed  background of  both plants as well as their medicinal applications including their use in combating sickle cell disease. The methods are  described in great detail with very comprehensive data  presented and discussed showing the diversity of their phytochemicals. Of particular note are the docking studies that demonstrated the ability of  vernodalol, an elemenolide-type sesquiterpene lactone  from VAL plus a related new compound, lasiopulide, reported  for the first time. Of particular importance is their efficacy against  sickle-cell disease due to their affinity for P-selectin and deoxyhemogobin. Both plants appeared effective against  sickle-cell disease although there were slight differences between VAL and OCL  Based on the data  presented this  is  a seminal study that will lead to further work and the beneficial treatment of sickle-cell disease.    

Author Response

This is an outstanding paper that provides a very detailed analysis of  the extracts from  two  Nigerian medicinal plants, Vernonia amygdalina  (VAL) and Ocvimum gatissimum (OGL) that are able to survive not only drought and harsh conditions but also  crude oil spillage. The abstract  summarizes the  detailed work well. The introduction provides a detailed  background of  both plants as well as their medicinal applications including their use in combating sickle cell disease. The methods are  described in great detail with very comprehensive data  presented and discussed showing the diversity of their phytochemicals. Of particular note are the docking studies that demonstrated the ability of  vernodalol, an elemenolide-type sesquiterpene lactone  from VAL plus a related new compound, lasiopulide, reported  for the first time. Of particular importance is their efficacy against  sickle-cell disease due to their affinity for P-selectin and deoxyhemogobin. Both plants appeared effective against  sickle-cell disease although there were slight differences between VAL and OCL  Based on the data  presented this  is  a seminal study that will lead to further work and the beneficial treatment of sickle-cell disease

Authors’ response:  We sincerely thank Reviewer 2 for taking the time to read the manuscript and come up with evidence-based comments carefully. Our findings have certainly laid a good background for a more detailed and mechanistic scientific investigation of the medicinal plants widely used in Nigeria.

Reviewer 3 Report

This is an interesting paper evaluating the similarities and differences in bioactivity and phytochemicals between VAL and OGL collected from the crude oil-rich Kokori and VAL and OGL collected from the adjacent crude oil-free Abraka. The work provides a reference for the use of mass spectrometry and metabolomics methods to study the changes of plant effective components in different cultivation environment.

Some minor modifications suggested but not required are as follows:

There are differences in harvest time and maturity for the samples in Table 1. How did the authors characterize these differences?

Line 245. Please re-edit the formula with the formula editor.

Line 250. Section 2.8 is placed at the end of the experimental section.

Table 3 is too large and is recommended to be placed in supporting materials. It is recommended to provide a typical UPLC-MS spectrum with the main peak numbers marked to give readers a general understanding. Values for the relative amounts of compounds are also suggested to be given in the supporting material.

Line 453. What does a line of words mean, subtitle?

What is the ordinate y in Figure 8. Errors and statistical comparisons are suggested for the data in the figures.

Line 517. It is suggested to add a sentence and recent references to bridge the molecular modeling study. For example, “Recently, molecular modeling methods have been widely used to illustrate the interactions between proteins and small molecules [Molecules 26(19), (2021), 5855; Food Chemistry 391 (2022) 133288].”

Line 534. It is recommended to convert the unit of kcal/mol to kJ/mol.

It is suggested to discuss the hydrophobic-hydrophobic interaction mode and hydrogen bonding interaction mode in Figures 9 and 10.

Line 612. Relevant reference and citation [123] are suggested to be placed in the Discussion section.

Please simplify the conclusion section.

Author Response

  1. 1. There are differences in harvest time and maturity for the samples in Table 1. How did the authors characterize these differences?

Authors’ response: In agreement with reviewer’s comment, authors have now added the following important text to the section (line number 151) preceding Table 1: “The plant samples were collected in April, 2017 during the dry season just before the onset of the raining season”. This is to ensure reproducibility of results during future studies as authors agree that environment and seasonal variations could significantly affect the accumulation of secondary metabolites in plants. This revision has also been added to Table 1  (line number 150) to make it similarly self-explanatory.

  1. 2. Line 245. Please re-edit the formula with the formula editor.

Authors’ response: Formula has been edited as suggested( line number 240-244).

  1. 3. Line 250. Section 2.8 is placed at the end of the experimental section.

Authors’ response: Section 2.8 has been repositioned as suggested. It is now section 2.11 in the revised manuscript (line number 285).

  1. 4. Table 3 is too large and is recommended to be placed in supporting materials. It is recommended to provide a typical UPLC-MS spectrum with the main peak numbers marked to give readers a general understanding. Values for the relative amounts of compounds are also suggested to be given in the supporting material.

Author response: We have moved Table 3  to the supplementary file as suggested. For the UPLC-MS base peak chromatogram numbering, we did provide retention time for each dereplicated base peak and other information detailed in Table S3. For the relative amounts of compounds, we appreciate your suggestion. However, the aim of the research is not to quantify the amounts of compounds in each base peak using UPLC-MS.

  1. 5. Line 453. What does a line of words mean, subtitle?

Authors’ response: We could not find the word “subtitle’’ in line 453 of the original manuscript.

  1. 6. What is the ordinate y in Figure 8. Errors and statistical comparisons are suggested for the data in the figures.

Authors’ response: In agreement with reviewer’s comment, authors have added more information to clearly explain this section by providing meaning to the ordinate y and included errors for the data in Table 3 and Figure 6 ( please see line numbers 509-519)

  1. 7. Line 517. It is suggested to add a sentence and recent references to bridge the molecular modeling study. For example, “Recently, molecular modeling methods have been widely used to illustrate the interactions between proteins and small molecules [Molecules 26(19), (2021), 5855; Food Chemistry 391 (2022) 133288].”

Authors’ response: Thank you for the  suggestion. This has been included in the revised manuscript (line number 534)

  1. 8. Line 534. It is recommended to convert the unit of kcal/mol to kJ/mol.

Authors’ response: The unit has been converted, and the new values reflected in the revised manuscript(line number 584). We appreciate the reviewer’s correction.

  1. 9. It is suggested to discuss the hydrophobic-hydrophobic interaction mode and hydrogen bonding interaction mode in Figures 9 and 10.

Author response: This information has now been provided in the revised manuscript (line numbers 569-582). We appreciate your suggestions.

  1. 10. Line 612. Relevant reference and citation [123] are suggested to be placed in the Discussion section.

Authors’ response: Thank you for your suggestion. This has now been addressed.

  1. 11. Please simplify the conclusion section.

Author response: The authors have carefully revised the key findings in the conclusion (line number 650-670)
